# Graph Neural Networks are Dynamic Programmers

**Andrew Dudzik**∗
DeepMind
adudzik@deepmind.com

**Petar Veličković**∗
DeepMind
petarv@deepmind.com

## Abstract

Recent advances in neural algorithmic reasoning with graph neural networks (GNNs) are propped up by the notion of algorithmic alignment. Broadly, a neural network will be better at learning to execute a reasoning task (in terms of sample complexity) if its individual components align well with the target algorithm. Specifically, GNNs are claimed to align with dynamic programming (DP), a general problem-solving strategy which expresses many polynomial-time algorithms. However, has this alignment truly been demonstrated and theoretically quantified? Here we show, using methods from category theory and abstract algebra, that there exists an intricate connection between GNNs and DP, going well beyond the initial observations over individual algorithms such as Bellman-Ford. Exposing this connection, we easily verify several prior findings in the literature, produce better-grounded GNN architectures for edge-centric tasks, and demonstrate empirical results on the CLRS algorithmic reasoning benchmark. We hope our exposition will serve as a foundation for building stronger algorithmically aligned GNNs.

## 1 Introduction

One of the principal pillars of neural algorithmic reasoning [27] is training neural networks that *execute* algorithmic computation in a high-dimensional latent space. While this process is in itself insightful, and can lead to stronger combinatorial optimisation systems [21], it is valuable in terms of expanding the applicability of classical algorithms. Evidence of this value are emerging, with pre-trained algorithmic reasoners utilised in implicit planning [11] and self-supervised learning [28].

A fundamental question in this space is: which *architecture* should be used to learn a particular algorithm (or collection of algorithms [36])? Naturally, we seek architectures that have low *sample complexity*, as they will allow us to create models that generalise better with fewer training examples.

The key theoretical advance towards achieving this aim has been made by [37]. Therein, the authors formalise the notion of *algorithmic alignment*, which states that we should favour architectures that align better to the algorithm, in the sense that we can separate them into modules, which individually correspond to the computations of the target algorithm's subroutines. It can be proved that architectures with higher algorithmic alignment will have lower sample complexity in the NTK regime [20]. Further, the theory of [37] predicts that graph neural networks (GNNs) algorithmically align with dynamic programming [3, DP]. The authors demonstrate this by forming an analogy to the Bellman-Ford algorithm [2].

Since DP is a very general class of problem-solving techniques that can be used to express many classical algorithms, this finding has placed GNNs as the central methodology for neural algorithmic execution [7]. However, it quickly became apparent that it is not enough to just train any GNN—for many algorithmic tasks, careful attention is required. Several papers illustrated special cases of GNNs that align with sequential algorithms [31], linearithmic sequence processing [16], physics simulations

---

∗Equal contribution.

[23], iterative algorihtms [26], data structures [29] or auxiliary memory [24]. Some explanations for this lack of easy generalisation have arisen—we now have both geometric [38] and causal [4] views into how better generalisation can be achieved.

We believe that the fundamental reason why so many isolated efforts needed to look into learning specific classes of algorithms is the fact the GNN-DP connection *has not been sufficiently explored*. Indeed, the original work of [37] merely mentions in passing that the formulation of DP algorithms seems to align with GNNs, and demonstrates one example (Bellman-Ford). Our thorough investigation of the literature yielded no concrete follow-up to this initial claim. But DP algorithms are very rich and diverse, often requiring a broad spectrum of computations. Hence what we really need is a *framework* that could allow us to identify GNNs that could align particularly well with certain *classes* of DP, rather than assuming a "one-size-fits-all" GNN architecture will exist.

As a first step towards this, in this paper we interpret the operations of *both* DP and GNNs from the lens of category theory and abstract algebra. We elucidate the GNN-DP connection by observing a diagrammatic abstraction of their computations, recasting algorithmic alignment to aligning the diagrams of (G)NNs to ones of the target algorithm class. In doing so, several previously shown results will naturally arise as corollaries, and we propose novel GNN variants that empirically align better to edge-centric algorithms. We hope our work opens up the door to a broader unification between algorithmic reasoning and the geometric deep learning blueprint [5].

## 2   GNNs, dynamic programming, and the categorical connection

Before diving into the theory behind our connection, we provide a quick recap on the methods being connected: graph neural networks and dynamic programming. Further, we cite related work to outline why it is sufficient to interpret DP from the lens of graph algorithms.

We will use the definition of GNNs based on [5]. Let a graph be a tuple of *nodes* and *edges*, $G = (V, E)$, with one-hop neighbourhoods defined as $\mathcal{N}_u = \{v \in V \mid (v, u) \in E\}$. Further, a node feature matrix $\mathbf{X} \in \mathbb{R}^{|V| \times k}$ gives the features of node $u$ as $\mathbf{x}_u$; we omit edge- and graph-level features for clarity. A *(message passing)* GNN over this graph is then executed as:

$$\mathbf{h}_u = \phi\left(\mathbf{x}_u, \bigoplus_{v \in \mathcal{N}_u} \psi(\mathbf{x}_u, \mathbf{x}_v)\right) \tag{1}$$

where $\psi : \mathbb{R}^k \times \mathbb{R}^k \to \mathbb{R}^k$ is a *message function*, $\phi : \mathbb{R}^k \times \mathbb{R}^k \to \mathbb{R}^k$ is a *readout function*, and $\bigoplus$ is a permutation-invariant *aggregation function* (such as $\sum$ or $\max$). Both $\psi$ and $\phi$ can be realised as MLPs, but many special cases exist, giving rise to, e.g., attentional GNNs [30].

Dynamic programming is defined as a process that solves problems in a *divide et impera* fashion: imagine that we want to solve a problem instance $x$. DP proceeds to identify a set of *subproblems*, $\eta(x)$, such that solving them first, and recombining the answers, can directly lead to the solution for $x$: $f(x) = \rho(\{f(y) \mid y \in \eta(x)\})$. Eventually, we decompose the problem enough until we arrive at an instance for which the solution is trivially given (i.e. $f(y)$ which is known upfront). From these "base cases", we can gradually build up the solution for the problem instance we initially care for in a bottom-up fashion. This rule is often expressed programmatically:

$$\mathtt{dp[x]} \leftarrow \mathtt{recombine(score(dp[y], dp[x])\ for\ y\ in\ expand(x))} \tag{2}$$

To initiate our discussion on why DP can be connected with GNNs, it is a worthwhile exercise to show how Equation 2 induces a *graph* structure. To see this, we leverage a categorical analysis of dynamic programming first proposed by [10]. Therein, dynamic programming algorithms are reasoned about as a composition of *three* components (presented here on a high level):

$$\mathtt{dp} = \underbrace{\rho}_{\mathtt{recombine}} \circ \underbrace{\sigma}_{\mathtt{score}} \circ \underbrace{\eta}_{\mathtt{expand}} \tag{3}$$

Expansion selects the relevant subproblems; scoring computes the quality of each individual subproblem's solution w.r.t. the current problem, and recombining combines these solutions into a solution for the original problem (e.g. by taking the max, or average).

Therefore, we can actually identify every subproblem as a *node* in a *graph*. Let $V$ be the space of all subproblems, and $R$ an appropriate value space (e.g. the real numbers). Then, expansion is defined

as $\eta : V \to \mathcal{P}(V)$, giving the set of all subproblems relevant for a given problem. Note that this also induces a set of *edges* between subproblems, $E$; namely, $(x, y) \in E$ if $x \in \eta(y)$. Each subproblem is scored by using a function $\sigma : \mathcal{P}(V) \to \mathcal{P}(R)$. Finally, the individual scores are recombined using the recombination function, $\rho : \mathcal{P}(R) \to R$. The final dynamic programming primitive therefore computes a function dp $: V \to R$ in each of the subproblems of interest.

Therefore, dynamic programming algorithms can be seen as performing computations over a *graph of subproblems*, which can usually be precomputed for the task at hand (since the outputs of $\eta$ are assumed known upfront for every subproblem). One specific popular example is the Bellman-Ford algorithm [2], which computes single-source shortest paths from a given source node, $s$, in a graph $G = (V, E)$. In this case, the set of subproblems is exactly the set of nodes, $V$, and the expansion $\eta(u)$ is exactly the set of one-hop neighbours of $u$ in the graph. The algorithm maintains *distances* of every node to the source, $d_u$. The rule for iteratively recombining these distances is as follows:

$$d_u \quad \leftarrow \quad \min \left( d_u, \min_{v \in \mathcal{N}_u} d_v + w_{v \to u} \right) \tag{4}$$

where $w_{v \to u}$ is the distance between nodes $v$ and $u$. The algorithm's base cases are $d_s = 0$ for the source node, $d_u = +\infty$ otherwise. Note that more general forms of Bellman-Ford pathfinding exist, for appropriate definitions of + and $\min$ (in general known as a *semiring*). Several recent research papers such as NBFNet [39] explicitly call on this alignment in their motivation.

## 3    The difficulty of connecting GNNs and DP

The basic technical obstacle to establishing a rigorous correspondence between neural networks and DP is the vastly different character of the computations they perform. Neural networks are built from linear algebra over the familiar real numbers, while DP, which is often a generalisation of path-finding problems, typically takes place over "tropical" objects like $(\mathbb{N} \cup \{\infty\}, \min, +)^2$, which are usually studied in mathematics as "degenerations" of Euclidean space. The two worlds cannot clearly be reconciled, directly, with simple equations.

However, if we define an arbitrary "latent space" $R$ and make as few assumptions as possible, we can observe that many of the behaviors we care about, *for both GNNs and DP*, arise from looking at functions $S \to R$, where $S$ is a finite set. $R$ can be seen as the set of real-valued vectors in the case of GNNs, and the tropical numbers in the case of DP.

So our principal object of study is the category of finite sets, and "$R$-valued quantities" on it. By "category" here we mean a collection of *objects* (all finite sets) together with a notion of composable *arrows* (functions between finite sets).

To draw our GNN-DP connection, we need to devise an abstract object which can capture both the GNN's message passing/aggregation stages (Equation 1) and the DP's scoring/recombination stages (Equation 2). It may seem quite intuitive that these two concepts can and should be relatable, and category theory is a very attractive tool for *"making the obvious even more obvious"* [15]. Indeed, recently concepts from category theory have enabled the construction of powerful GNN architectures beyond permutation equivariance [9]. Here, we propose **integral transforms** as such an object.

We will construct the integral transform by composing transformations over our input features in a way that will depend minimally on the specific choice of $R$. In doing so, we will build a computational diagram that will be applicable for both GNNs and DP (and their own choices of $R$), and hence allowing for focusing on making components of those diagrams as aligned as possible.

## 4    The integral transform

An integral transform can be encoded in a diagram of this form, which we call a *polynomial span*:

---

[2]Here we require the addition of a special "$\infty$" placeholder object to denote the vertices the DP expansion hasn't reached so far.

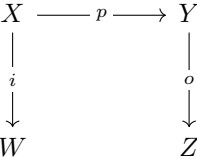

where $W, X, Y$ and $Z$ are finite sets. The arrows $i, p, o$ stand, respectively, for "input", "process", and "output". In context, the sets will have the following informal meaning: $W$ represents the set over which we define our *inputs*, $Z$ the set over which we define *outputs*. $X$ and $Y$ are, respectively, carrier sets for the *arguments*, and the *messages*[3]—we will clarify their meaning shortly.

Before proceeding, it is worthy to note the special case of $X = Y = E$, with $p$ being the identity map. Such a diagram is commonly known as a *span*. A span that additionally has $W = Z = V$ is equivalent to a representation of a directed graph with vertex set $Z$ and edge set $Y$ ($V \leftarrow E \rightarrow V$); in this case $i(e)$ and $o(e)$ are the functions identifying the source and target nodes of each edge.

The key question is: given input data $f$ on $W$, assigning features $f(w)$ to each $w \in W$, how to transform it, via the polynomial span, into data on $Z$? If we can do this, we will be able to characterise both the process of sending messages between nodes in GNNs *and* scoring subproblems in DP.

For us, data on a carrier set $S$ consists of an element of $[S, R] := \{f : S \rightarrow R\}$, where $R$ is a "set of possible values". For now, we will think of $R$ as an arbitrary (usually infinite) set, though we will see later that it should possess some algebraic structure; it should be a semiring.

The transform proceeds in three steps, following the edges of the polynomial span:

$$
\begin{array}{ccc}
[X, R] & \overset{p_\otimes}{\xrightarrow{\hspace{2cm}}} & [Y, R] \\
\uparrow{\scriptstyle i^*} & & \downarrow{\scriptstyle o_\oplus} \\
[W, R] & & [Z, R]
\end{array}
\tag{5}
$$

We call the three arrows $i^*, p_\otimes, o_\oplus$ the *pullback*, the *argument pushfoward*, and the *message pushforward*. Taken together, they form an *integral transform*—and we conjecture that this transform can be described as a polynomial functor, where $p_\otimes$ and $o_\oplus$ correspond to the *dependent product* and *dependent sum* from type theory (cf. Appendix D for details).

The pullback $i^*$ is the easiest to define. Since we have a function $i : X \rightarrow W$ (part of the polynomial span) and a function $f : W \rightarrow R$ (our input data), we can produce data on $X$, that is, a function in $X \rightarrow R$, by composition. We hence define $i^* f = f \circ i$.

Unfortunately, the other two arrows of the polynomial span point in the wrong direction for naïve composition. For the moment, we will focus on how to define $o_\oplus$ and leave $p_\otimes$ for later.

We start with message data $m : Y \rightarrow R$. It may be attractive to *invert* the output arrow $o$ in order to define a composition with $o^{-1}$, as was done in the case of the pullback. However, unless $o$ is bijective, the preimage $o^{-1} : Z \rightarrow \mathcal{P}(Y)$ takes values in the *power set* of $Y$. There is an additional technicality: if the composition $m \circ o^{-1}$ takes values in $\mathcal{P}(R)$, it will fail to detect multiplicities; we are unable to tell from a subset of $R$ whether multiple messages had the same value.

So instead, our pushforward takes values in $\texttt{bag}(R)$, the set of finite multisets (or bags) of $R$, which we describe in more detail in appendix B. For the moment, it is enough to know that a bag is equivalent to a formal sum, and we define an intermediate message pushforward $(\overline{o_\oplus}m)(u) := \Sigma_{e \in t^{-1}(u)} m(e) \in [Z, \texttt{bag}(R)]$.

---

[3]For technical reasons, we ask that for each $y \in Y$, the preimage $p^{-1}(y) = \{x \in X \mid p(x) = y\}$ should have a total order. This is to properly support functions with non-commuting arguments, though this detail is unnecessary for our key examples, where arguments commute.

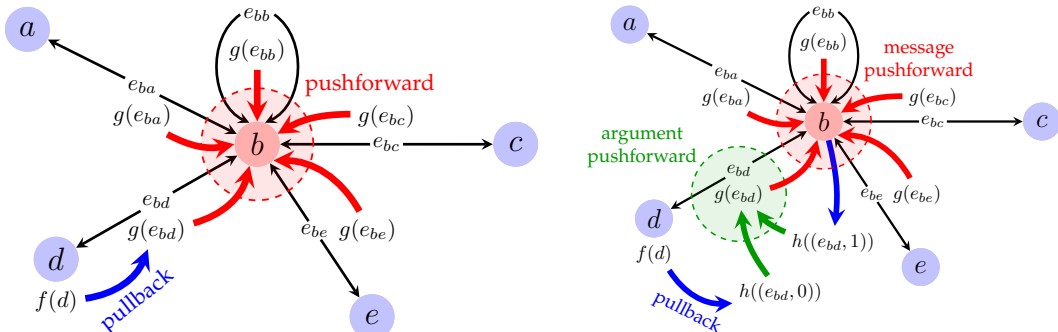

Figure 1: The illustration of how pullback and pushforward combine to form the *integral transform*, for two specific cases. **Left:** Polynomial span $V \leftarrow E \rightarrow E \rightarrow V$ with trivial argument pushforward (identity). Each edge $e_{uv}$ is connected to its sender and receiver nodes $(u, v)$ via the *span* (black arrows). The *pullback* then "pulls" the node features $f(u)$ along the span, which the *argument pushforward* folds into edge features $g(e_{vu}) = f(u)$. Once all sender features are pulled back to their edges, the *message pushforward* then "collects" all of the edge features that send to a particular receiver, by pushing them along the span. **Right:** Polynomial span $V \leftarrow E + E \rightarrow E \rightarrow V$, a situation more commonly found in GNNs. In this case, the pullback pulls sender and receiver node features into the argument function, $h$. The *argument pushforward* then computes, from these arguments, the edge messages, $g$, which are sent to receivers via the *message pushforward*, as before. See Appendix A for a visualisation of how these arrows translate into GNN code.

All that is missing to complete our definition of $o_\oplus$ is an *aggregator* $\bigoplus : \mathtt{bag}(R) \rightarrow R$. As we will see later, specifying a well-behaved aggregator is the same as imposing a *commutative monoid* structure on $R$. With such an aggregator on $R$, we can define $(o_\oplus m)(u) := \bigoplus(\overline{o_\oplus}m)(u)$.

We return to $p_\otimes$, which is constructed very similarly. The only difference is that, while we deliberately regard the collection of messages as unordered, the collection of arguments used to compute a message has an *ordering* we wish to respect. So instead of the type $\mathtt{bag}(R)$, we use the type $\mathtt{list}(R)$ of finite lists of elements of $R$, and our aggregator $\bigotimes : \mathtt{list}(R) \rightarrow R$ is now akin to a *fold* operator.

We illustrate the use of these two aggregators in a decomposed diagram:

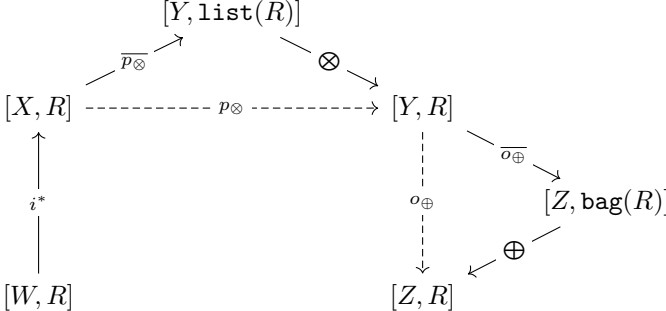

Note that any semiring $(R, \otimes, \oplus)$ comes equipped with binary operators $\otimes, \oplus$ that allow aggregators $\bigotimes, \bigoplus$ to be defined inductively. In fact, the converse—that every set with two such aggregators is a semiring—is also true, if we assume some reasonable conditions on the aggregators, which we can explain in terms of one of the most utilised concepts in category theory and functional programming—*monads* [33]. Due to space constraints, we refer the interested reader to Appendices B and C for a full exposition of how we can use monads over lists and bags to constrain the latent space $R$ to respect a semiring structure.

For now, it's enough to know that our key examples of the real numbers (with multiplication and addition, for GNNs) and the tropical natural numbers (with addition and minimum, for DP) both allow for natural interpretations of $\otimes$ and $\oplus$ in the integral transform.

We are now ready to show how the integral transform can be used to instantiate popular examples of algorithms and GNNs. We start with the Bellman-Ford algorithm [2] (Equation 4) that was traditionally used to demonstrate the concept of algorithmic alignment.

## 5 Bellman-Ford

Let $R = (\mathbb{N} \cup \{\infty\}, +, \min)$ be the "min-plus" semiring of extended natural numbers, with $\otimes = +$ and $\oplus = \min$. This is the coefficient semiring over which the Bellman-Ford algorithm takes place.

Let $(V, E)$ be a weighted graph with source and target maps $s, t : E \to V$ and edge weights $w : E \to R$. For purely technical reasons, we also need to explicitly materialise a *bias function* $b : V \to R$, which is, in practice, a constant-zero function ($b(v) = 0$ for all $v \in V$) but will prove necessary for defining the argument pushforward.

We interpret Bellman-Ford as the following polynomial span:

$$
\begin{array}{ccc}
(V + E) + (V + E) & \overset{p}{\longrightarrow} & V + E \\
\big\downarrow i & & \big\downarrow o \\
V + (V + E) & & V
\end{array}
\tag{6}
$$

Here "+" is the disjoint union of sets, defined as $A + B = \{(a, 1) \mid a \in A\} \cup \{(b, 2) \mid b \in B\}$. Note that $[S + T, R] \cong [S, R] \times [T, R]$, i.e. specifying data on a disjoint union is equivalent to specifying data on each component separately.

Initially, we describe each of the four sets of the polynomial span, making their role clear:

- **Input:** $W = V + (V + E)$. Our input to Bellman-Ford includes: the current estimate of node distances ($d_u$; a function in $[V, R]$), edge weights ($w$; a function in $[E, R]$), and the previously discussed bias $b$, a function in $[V, R]$. Hence our overall inputs are members of $[V, R] \times [E, R] \times [V, R] \cong [V + (V + E), R]$, justifying our choice of input space.

- **Arguments:** $X = (V + E) + (V + E)$. Here we collect the ingredients necessary to compute Bellman-Ford's subproblem solutions coming from neighbouring nodes. To do this, we need to combine data in the nodes with data living on edges—those are the *arguments* to the function. And since they meet in the edges, we "lift" our node distances $[V, R]$ to edges they are sending from, giving us an additional function in $[E, R]$. Hence our argument carrier space is now $(V + E) + (V + E)$ (the remaining three inputs remain unchanged).

- **Message:** $Y = V + E$. Once the arguments are combined to compute messages, we are left with signal in each edge (containing the sum of corresponding $d_u$ and $w_{uv}$), and each node (containing just $d_u$, for the purposes of access to the previous optimal solution). Hence our messages are members of $[V, R] \times [E, R]$, justifying our choice of message space.

- **Output:** $Z = V$. Lastly, the output of one step of Bellman-Ford are updated values $d'_u$, which we can interpret as just (output) data living on $V$.

We now describe how to propagate data along each arrow of the diagram in turn, beginning with inputs $(f, b, w)$ of node features $f : V \to R$, a bias $b : V \to R$, and edge weights $w : E \to R$:

- **Pullback,** $i^*$**:** First, we can note the input function $i : (V + E) + (V + E) \to V + (V + E)$ decomposes as the sum of two arrows. $i_1 : V + E \to V$ is the identity function on $V$ and the source function on $E$, and $i_2 : V + E \to V + E$ is just the identity. So we calculate the pullback $i^*(f, b, w) = (f, f \circ s, b, w)$, giving us the arguments to compute messages.

- **Argument pushforward,** $p_\otimes$**:** Next, the process function $p$ simply identifies the two copies of $V + E$, and sums their values. So the argument pushforward is $p_\otimes(f, f \circ s, b, w) =$

$(f, f \circ s) \otimes (b, w) = (f + b, (f \circ s) + w)$. This also allows us to interpret the bias function, $b$, as a "self-edge" in the graph with weight 0.

- **Message pushforward, $o_\oplus$:** The output function $o : V + E \to V$ is the identity function on $V$ and the target function on $E$. So the message pushforward gives us $(o_\oplus(f + b, (f \circ s) + w))(u) = (f(u) + b(u)) \oplus \bigoplus_{t(e)=u}(f \circ s)(e) = \min(f(u) + b(u), \min_{v \to u} f(v) + w_{v \to u})$.

Letting $b(u) = 0$, we can see that this is *exactly* Equation 4. So we have produced the formula for the Bellman-Ford algorithm directly from the polynomial span in Diagram 6.

Note that $p_\oplus$ is aligned with using max aggregation in neural networks—directly explaining several previous proposals, such as [31]. But additionally, $p_\otimes$, as defined, is aligned with concatenating all message arguments together and passing them through a linear function, which is how such a step is implemented in GNNs' message functions. We now direct our polynomial span analysis at GNNs.

## 6  GNNs

We study the popular *message passing neural network* (MPNN) model [19], which can be interpreted using the following polynomial span diagram:

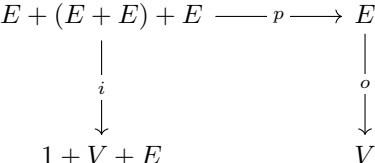

Here the set $1$ refers to a singleton set—sometimes also called $()$, or *unit*—which is used as a carrier for graph-level features. This implies the graph features will be specified as $[1, R] \cong R$, as expected.

Given all these features, how would we compute messages? The natural way is to combine the features of the sender and receiver node of each edge, features of said edge, and graph-level features—these will form our arguments, and they need to all "meet" in the edges. This motivates our argument space as $E + (E + E) + E$: all of the above four, accordingly broadcast into their respective edge(s).

The input map, $i$, is then the unique map to the singleton, the sender and receiver functions on the two middle copies of $E$, and the identity on the last copy of $E$, i.e. $i(a, b, c, d) = \{(), s(b), t(c), d\}$. The process map, $p$, collapses the four copies of $E$ into just one, to hold the computed message. Lastly, the output map, $o$, is the target function, identifying the node to which the message will be delivered.

The actual computation performed by the network (over real values in $\mathbb{R}$, which can support various semirings of interest) is exactly an integral transform, with an extra MLP processing step on messages:

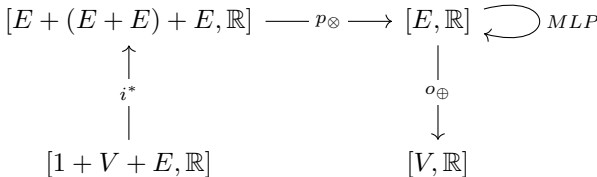

It is useful to take a moment to discuss what was just achieved: with a single abstract template (the polynomial span), we have successfully explained both a dynamic programming algorithm, and a GNN update rule—merely by choosing the correct support sets and latent space.

## 7  Improving GNNs with edge updates, with experimental evaluation

From now on, we will set $E = V^2$, as all our baseline GNNs will use fully connected graphs, and it will accentuate the *polynomial* nature of our construction.

We now show how our polynomial span view can be used to directly propose better-aligned GNN architectures for certain algorithmic tasks. Since the MPNN diagram above outputs only node features, to improve predictive performance on edge-centric algorithms, it is a natural augmentation to also update edge features, by adding edges to the output carrier (as done by, e.g., [1]):

$$
\begin{array}{ccc}
V^2 + (V^2 + V^2) + V^2 & \xrightarrow{\quad p \quad} & V^2 \\
\downarrow{\scriptstyle i} & & \downarrow{\scriptstyle o} \\
1 + V + V^2 & & V + V^2
\end{array}
\tag{7}
$$

But notice that there is a problem with the output arrow. Since we are using each message twice, $o$ is no longer a function—it'd have to send each edge message to *two* different objects! To resolve this, we need to appropriately augment the messages and the arguments. This is equivalent to specifying a new polynomial span with output $V^2$, which we can then recombine with Diagram 7:

$$
\begin{array}{ccc}
? & \xrightarrow{\quad p \quad} & ? \\
\downarrow{\scriptstyle i} & & \downarrow{\scriptstyle o} \\
1 + V + V^2 & & V^2
\end{array}
\tag{8}
$$

Most edge-centric algorithms of interest (such as the Floyd-Warshall algorithm for all-pairs shortest paths [14]), compute edge-level outputs by *reducing* over a choice of "intermediate" node. Hence, it would be beneficial to produce messages with shape $V^3$, which would then reduce to features over $V^2$. There are three possible ways to broadcast both node and edge features into $V^3$, so we propose the following polynomial span, which materialises each of those arguments:

$$
\begin{array}{ccc}
V^3 + (V^3 + V^3 + V^3) + (V^3 + V^3 + V^3) & \xrightarrow{\quad p \quad} & V^3 \\
\downarrow{\scriptstyle i} & & \downarrow{\scriptstyle o} \\
1 + V + V^2 & & V^2
\end{array}
\tag{9}
$$

Finally, inserting this into Diagram 7 gives us a corrected polynomial span with output $V + V^2$:

$$
\begin{array}{ccc}
4V^2 + 7V^3 & \xrightarrow{\quad p \quad} & V^2 + V^3 \\
\downarrow{\scriptstyle i} & & \downarrow{\scriptstyle o} \\
1 + V + V^2 & & V + V^2
\end{array}
\tag{10}
$$

Here we have collapsed the copies of $V^2$ and $V^3$ in the argument position for compactness.

While Diagram 7 doesn't make sense as a polynomial diagram of sets, we can clearly still implement it as an architecture [1], since nothing stops us from sending the same tensor to two places. We want to investigate whether our proposed modification of Diagram 10, which materialises order-3 messages, leads to improved algorithmic alignment on edge-centric algorithms. To support this evaluation, we initially use a set of six tasks from the recently proposed CLRS Algorithmic Reasoning Benchmark [32], which evaluates how well various (G)NNs align to classical algorithms, both in- and out-of-distribution. We reuse exactly the data generation and base model implementations in the publicly available code for the CLRS benchmark.

We implemented each of these options by making our GNN's message and update functions be two-layer MLPs with embedding dimension 24, and hidden layers of size 8 and 16. Our test results (out-of-distribution) are summarised in Table 1. For convenience, we also illustrate the in-distribution performance of our models via plots given in Appendix E.

Lastly, we scale up our experiments to 27 different tasks in CLRS, 96-dimensional embeddings, and using the PGN processor [29], which is the current state-of-the-art model on CLRS in terms of task win count [32]. We summarise the performance improvement obtained by our $V^3$ variant of PGN in Table 2, aggregated across edge-centric tasks as well as ones that do not require explicit edge-level reasoning. For convenience, we provide the per-task test performance in Appendix F (Table 3).

We found that the $V^3$ architecture was equivalent to, or outperformed, the non-polynomial ($V^2$) one in all edge-centric algorithms (up to standard error). Additionally, this architecture appears to also provide some gains on tasks without explicit edge-level reasoning requirements, albeit smaller on average and less consistently. Our result directly validates our theory's predictions, in the context of presenting a better-aligned GNN for edge-centric algorithmic targets.

Table 1: Test (out-of-distribution) results of our models on all models on the six algorithms studied. $V^2$ corresponds to the baseline model offered by Diagram 7, while $V^3$ corresponds to our proposal in Diagram 10, which respects the polynomial span.

| Algorithm | $V^2$-large | $V^3$-large | $V^2$-small | $V^3$-small |
|---|---|---|---|---|
| Dijkstra | $59.58\% \pm 2.82$ | $\mathbf{68.53}\% \pm \mathbf{2.40}$ | $56.10\% \pm 3.25$ | $60.32\% \pm 2.70$ |
| Find Maximum Subarray | $8.33\% \pm 0.50$ | $\mathbf{9.06}\% \pm \mathbf{0.65}$ | $8.46\% \pm 0.55$ | $7.89\% \pm 0.64$ |
| Floyd-Warshall | $7.46\% \pm 0.63$ | $\mathbf{9.00}\% \pm \mathbf{0.81}$ | $6.66\% \pm 0.62$ | $8.23\% \pm 0.62$ |
| Insertion Sort | $15.39\% \pm 1.27$ | $\mathbf{24.67}\% \pm \mathbf{2.44}$ | $14.69\% \pm 1.32$ | $20.23\% \pm 2.21$ |
| Matrix Chain Order | $67.64\% \pm 1.23$ | $\mathbf{70.79}\% \pm \mathbf{1.54}$ | $68.85\% \pm 2.26$ | $68.76\% \pm 1.21$ |
| Optimal BST | $53.03\% \pm 2.80$ | $\mathbf{54.56}\% \pm \mathbf{4.34}$ | $46.65\% \pm 3.82$ | $51.94\% \pm 4.60$ |
| Overall average | $35.24\%$ | $\mathbf{39.43}\%$ | $33.57\%$ | $36.23\%$ |

Table 2: Test (out-of-distribution) results across 27 tasks in CLRS, for the PGN processor network, averaged across edge-centric and other tasks. See Appendix F for the per-task test performances.

| Algorithms | $V^2$–PGN | $V^3$–PGN | Average Improvement |
|---|---|---|---|
| Edge-centric algorithms | $35.03\%$ | $\mathbf{39.08}\%$ | $4.44\% \pm 1.06$ |
| Other algorithms | $35.37\%$ | $\mathbf{36.33}\%$ | $1.01\% \pm 0.11$ |
| Average of the two groups | $35.20\%$ | $\mathbf{37.70}\%$ | $2.73\%$ |

## 8   Conclusions

In this paper, we describe the use of category theory and abstract algebra to explicitly expand on the GNN-DP connection, which was previously largely handwaved on specific examples. We derived a generic diagram of an *integral transform* (based on standard categorical concepts like pullback, pushforward and commutative monoids), and argued why it is general enough to support both GNN and DP computations. With this diagram materialised, we were able to immediately unify large quantities of prior work as simply manipulating one arrow or element in the integral transform. We also provided empirical evidence of the utility of polynomial spans for analysing GNN architectures, especially in terms of algorithmic alignment. It is our hope that our findings inspire future research into better-aligned neural algorithmic reasoners, especially focusing on generalising or diving into several aspects of this diagram.

Lastly, it is not at all unlikely that analyses similar to ours have already been used to describe other fields of science—beyond algorithmic reasoners. The principal ideas of *span* and *integral transform* are central to defining Fourier series [35], and appear in the analysis of Yang-Mills equations in particle physics [13]. Properly understanding the common ground behind all of these definitions may, in the very least, lead to interesting connections, and a shared understanding between the various fields they span.

## Acknowledgments and Disclosure of Funding

We would like to thank Charles Blundell, Tai-Danae Bradley, Taco Cohen, Bruno Gavranović, Bogdan Georgiev, Razvan Pascanu, Karolis Špukas, Grzegorz Świrszcz, and Vincent Wang-Maścianica for the very useful discussions and feedback on prior versions of this work.

Special thanks to Tamara von Glehn for key comments helping us to formally connect integral transforms to polynomial functors.

This research was funded by DeepMind.

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
