```
Require:
    Node features X ∈ ℝ^{n×k},
    Message function ψ : ℝ^k × ℝ^k → ℝ^m,
    Update function φ : ℝ^m → ℝ^m
Ensure: Latent features H ∈ ℝ^{n×m}
    Arg^snd ← tile(X, 0, n); // Arg^snd ∈ ℝ^{n×n×k}
    Arg^rcv ← tile(X, 1, n); // Arg^rcv ∈ ℝ^{n×n×k}
    for (u, v) ∈ V × V do
        msg_uv ← ψ(arg_u^snd, arg_v^rcv); // Msg ∈ ℝ^{n×n×m}
    end for
    for u ∈ V do
        h_u ← φ(⊕_{v∈V} msg_vu);
    end for
```

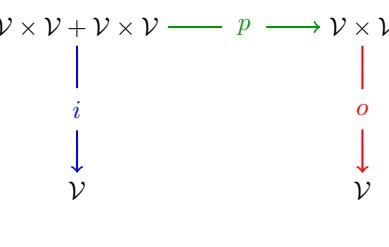

Figure 2: Correspondence between the individual arrows in the polynomial span, and the pseudocode steps for implementing a plausible graph neural network. The code sections are colour-coded to correspond to arrows in the polynomial span diagram. The specific sets $X, Y, V, W$ of the polynomial span are initialised to match the design choices in the GNN (for a set of nodes $\mathcal{V}$, such that $|\mathcal{V}| = n$).

## A   Correspondence between GNN pseudocode and the polynomial span

In Figure 2, we further elaborate on the diagrams given in Figure 1, to explicitly relate the various steps of how processing data with a GNN might proceed with the individual arrows $(i, p, o)$ of the polynomial span. To do this, we colour-code parts of a plausible GNN pseudocode, to match the colours of arrows in a polynomial span diagram.

Additionally, in Figure 3, we follow this construction to explicitly provide the pseudocodes for the proposed $V^2$ and $V^3$-GNN models (as proposed in Diagram 7 and Diagram 10, respectively).

## B   The bag and list monads

Before we conclude, we turn back to the theory behind our polynomial spans, to more precisely determine the restrictions on our abstract latent space $R$. We found this investigation useful to include in the main paper, as it yields a strong connection to one of the most actively used concepts in theoretical computer science and functional programming.

Recall that the realisation of our pushforward operations required the existence of two aggregators: $\otimes$ (to fold lists) and $\oplus$ (to reduce bags). Previously, we mentioned only in passing how they can be recovered—now, we proceed to define $\oplus$ axiomatically.

Given a set $S$, we define $\texttt{bag}(S) := \{p : S \to \mathbb{N} \mid \#\{p(r) \neq 0\} < \infty\}$, the natural-valued functions of finite support on $S$. This has a clear correspondence to multisets over $S$: $p$ sends each element of $S$ to the amount of times it appears in the multiset. We can write its elements formally as $\sum_{s \in S} n_s s$, where all but finitely many of the $n_s$ are nonzero.

Given a function $f : S \to T$ between sets, we can define a function $\texttt{bag}(f) : \texttt{bag}(S) \to \texttt{bag}(T)$, as follows: $\texttt{bag}(f)(\sum_{s \in S} n_s s) := \sum_{s \in S} n_s f(s)$, which we can write as $\sum_{t \in T} m_t t$, where $m_t = \sum_{f(s)=t} n_s$.

For each $S$, we can also define two special functions. The first is $\texttt{unit} : S \to \texttt{bag}(S)$, sending each element to its indicator function (i.e. an element $x \in S$ to the multiset $\{\!\{x\}\!\}$). The second is $\texttt{join} : \texttt{bag}(\texttt{bag}(S)) \to \texttt{bag}(S)$, which interprets a nested sum as a single sum.

These facts tell us that $\texttt{bag}$ is a **monad**, a special kind of self-transformation of the category of sets. Monads are very general tools for computation, used heavily in functional programming languages (e.g. Haskell) to model the semantics of wrapped or enriched types. Monads provide a clean way for abstracting control flow, as well as gracefully handling functions with *side effects* [33].

It is well-known that the algebras for the monad $\texttt{bag}$ are the *commutative monoids*, sets equipped with a commutative and associative binary operation and a unit element.

**Require:**
  Node features $\mathbf{X} \in \mathbb{R}^{n \times k}$,
  Message function $\psi : \mathbb{R}^k \times \mathbb{R}^k \to \mathbb{R}^m$,
  Update function $\phi : \mathbb{R}^m \to \mathbb{R}^m$
**Ensure:** Latent features $\mathbf{H} \in \mathbb{R}^{n \times m}$ (*nodes*), $\mathbf{M} \in \mathbb{R}^{n \times n \times m}$ (*edges*)
  $\mathbf{Arg}^{\mathtt{snd}} \leftarrow \mathtt{tile}(\mathbf{X}, 0, n); // \mathbf{Arg}^{\mathtt{snd}} \in \mathbb{R}^{n \times n \times k}$
  $\mathbf{Arg}^{\mathtt{rcv}} \leftarrow \mathtt{tile}(\mathbf{X}, 1, n); // \mathbf{Arg}^{\mathtt{rcv}} \in \mathbb{R}^{n \times n \times k}$
  **for** $(u, v) \in \mathcal{V} \times \mathcal{V}$ **do**
      $\mathbf{msg}_{uv} \leftarrow \psi(\mathbf{arg}_u^{\mathtt{snd}}, \mathbf{arg}_v^{\mathtt{rcv}}); // \mathbf{Msg} \in \mathbb{R}^{n \times n \times m}$
  **end for**
  **for** $u \in \mathcal{V}$ **do**
      $\mathbf{h}_u \leftarrow \phi(\bigoplus_{v \in \mathcal{V}} \mathbf{msg}_{vu});$
  **end for**
  $\mathbf{M} \leftarrow \mathbf{Msg}$
  // $\mathbf{Msg}$ is sent to two places ($\mathbf{H}, \mathbf{M}$); output morphism $o$ is not a function!

**Require:**
  Node features $\mathbf{X} \in \mathbb{R}^{n \times k}$,
  Edge message function $\psi^{(e)} : \mathbb{R}^k \times \mathbb{R}^k \to \mathbb{R}^m$,
  Triplet message function $\psi^{(t)} : \mathbb{R}^k \times \mathbb{R}^k \times \mathbb{R}^k \to \mathbb{R}^m$,
  Node update function $\phi^{(n)} : \mathbb{R}^m \to \mathbb{R}^m$,
  Edge update function $\phi^{(e)} : \mathbb{R}^m \to \mathbb{R}^m$
**Ensure:** Latent features $\mathbf{H} \in \mathbb{R}^{n \times m}$ (*nodes*), $\mathbf{M} \in \mathbb{R}^{n \times n \times m}$ (*edges*)
  $\mathbf{Arg}^{\mathtt{snd}} \leftarrow \mathtt{tile}(\mathbf{X}, 0, n); // \mathbf{Arg}^{\mathtt{snd}} \in \mathbb{R}^{n \times n \times k}$
  $\mathbf{Arg}^{\mathtt{rcv}} \leftarrow \mathtt{tile}(\mathbf{X}, 1, n); // \mathbf{Arg}^{\mathtt{rcv}} \in \mathbb{R}^{n \times n \times k}$
  $\mathbf{Arg}^{\mathtt{tri1}} \leftarrow \mathtt{tile}(\mathbf{X}, [0, 1], n); // \mathbf{Arg}^{\mathtt{tri1}} \in \mathbb{R}^{n \times n \times n \times k}$
  $\mathbf{Arg}^{\mathtt{tri2}} \leftarrow \mathtt{tile}(\mathbf{X}, [0, 2], n); // \mathbf{Arg}^{\mathtt{tri2}} \in \mathbb{R}^{n \times n \times n \times k}$
  $\mathbf{Arg}^{\mathtt{tri3}} \leftarrow \mathtt{tile}(\mathbf{X}, [1, 2], n); // \mathbf{Arg}^{\mathtt{tri3}} \in \mathbb{R}^{n \times n \times n \times k}$
  **for** $(u, v) \in \mathcal{V} \times \mathcal{V}$ **do**
      $\mathbf{msg}_{uv}^{\mathtt{edge}} \leftarrow \psi^{(e)}(\mathbf{arg}_u^{\mathtt{snd}}, \mathbf{arg}_v^{\mathtt{rcv}}); // \mathbf{Msg}^{\mathtt{edge}} \in \mathbb{R}^{n \times n \times m}$
      **for** $w \in \mathcal{V}$ **do**
          $\mathbf{msg}_{uvw}^{\mathtt{tri}} \leftarrow \psi^{(t)}(\mathbf{arg}_u^{\mathtt{tri1}}, \mathbf{arg}_v^{\mathtt{tri2}}, \mathbf{arg}_w^{\mathtt{tri3}}); // \mathbf{Msg}^{\mathtt{tri}} \in \mathbb{R}^{n \times n \times n \times m}$
      **end for**
  **end for**
  **for** $u \in \mathcal{V}$ **do**
      $\mathbf{h}_u \leftarrow \phi^{(n)}(\bigoplus_{v \in \mathcal{V}} \mathbf{msg}_{vu}^{\mathtt{edge}});$
      **for** $v \in \mathcal{V}$ **do**
          $\mathbf{m}_{uv} \leftarrow \phi^{(e)}(\bigoplus_{w \in \mathcal{V}} \mathbf{msg}_{uvw}^{\mathtt{tri}});$
      **end for**
  **end for**

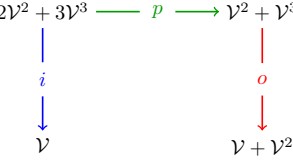

Figure 3: Correspondence between the arrows in the polynomial span, and the pseudocode for implementing the GNNs represented by Diagram 7 (**above**) and Diagram 10 (**below**). Edge and graph features are ignored for simpicity. The code sections are colour-coded to correspond to arrows in the polynomial span. Note the difference to Figure 2: we now also need to output edge features (on $\mathcal{V}^2$).

Concretely, a commutative monoid structure on a set $R$ is *equivalent* to defining an *aggregator* function $\bigoplus : \mathtt{bag}(R) \to R$ compatible with the unit and monad composition. Here, compatibility implies it should correctly handle sums of singletons and sums of sums, in the sense that the following two diagrams *commute*; that is, they yield the same result regardless of which path is taken:

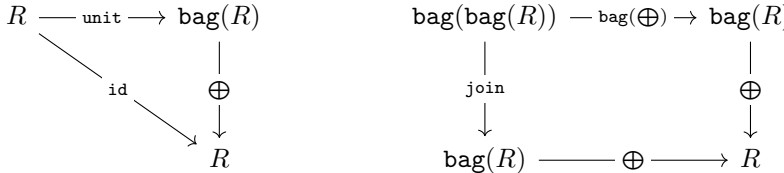

The first diagram explains that the outcome of aggregating a singleton multiset (i.e. the one produced by applying $\mathtt{unit}$) with $\bigoplus$ is equivalent to the original value placed in the singleton. The second diagram indicates that the $\bigoplus$ operator yields the same results over a nested multiset, regardless of whether we choose to directly apply it twice (once on each level of nesting), or first perform the $\mathtt{join}$ function to collapse the nested multiset, then aggregate the collapsed multiset with $\bigoplus$.

So the structure of a commutative monoid on $R$ is exactly what we need to complete our definition of the message pushforward $o_\oplus$. The story for the argument pushforward, $p_\otimes$, is remarkably similar.

Define $\texttt{list}(S) := \{(s_1, \ldots, s_n) \mid n \in \mathbb{N}, s_i \in S\}$, the set of all ordered lists of elements of $S$, including the empty list. Equivalently, $\texttt{list}(S) = \coprod_{n \geq 0} S^n$. We can also extend $\texttt{list}$ to a functor: for a function $f : S \to T$, $\texttt{list}(f) : \texttt{list}(S) \to \texttt{list}(T)$ is just the well-known $\texttt{map}$ operation: $\texttt{list}(f)(s_1, \ldots, s_n) := (f(s_1), \ldots, f(s_n))$.

$\texttt{list}$ is also a monad, with $\texttt{unit} : S \to \texttt{list}(S)$ sending each $x \in S$ to the singleton list $(x)$, and $\texttt{join} : \texttt{list}(\texttt{list}(S)) \to \texttt{list}(S)$ sending a list of lists to their concatenation.

The algebras for the list monads are *monoids*—not just commutative ones. So $R$ needs a second monoid structure, possibly noncommutative, to support our definition of the argument pushforward. We detail how this can elegantly be done in our specific case in Appendix C.

# C   The monad for semirings

We have asked that $R$ be an algebra for two monads: $\texttt{list}$ and $\texttt{bag}$. But this is an unnatural condition without some compatibility between the two. It would more useful to find a single monad encapsulating both.

In general, the composition of two monads is not a monad. For example, the composite functor $\texttt{list} \circ \texttt{bag}$ does not support a monad structure.

However, the other composite $\texttt{bag} \circ \texttt{list}$ is actually a monad in a natural way, due to the existence of a *distributive law*, which is a natural transformation $\lambda : \texttt{list} \circ \texttt{bag} \to \texttt{bag} \circ \texttt{list}$ satisfying some axioms, see e.g. [8].[4]

It is easy to describe $\lambda$. Given any list of bags $(\sum_{i_1} a_{i_1}, \ldots, \sum_{i_n} a_{i_n})$, we have $\lambda(\sum_{i_1} a_{i_1}, \ldots, \sum_{i_n} a_{i_n}) = \sum_{i_1, \ldots, i_n} (a_{i_1}, \ldots, a_{i_n})$. In other words, $\lambda$ takes a list of bags and returns the bag of all ordered selections from the list.

This is exactly how multiplication of sums works in a semiring. For example, if I think of a polynomial as a bag of monomials, and I want to compute a product of polynomials, I interpret this product as a list of polynomials, i.e. a list of bags. Then I expand it into a bag of lists (a sum of products), and finally perform the products to produce the resulting bag of monomials, i.e. polynomial.

So it shouldn't be a surprise that the algebras for the composite monad $\texttt{bag} \circ \texttt{list}$ are exactly semirings, i.e. sets $R$ equipped with a commutative monoid structure $\bigoplus$, another monoid structure $\bigotimes$, and a "distributive law" $\bigotimes \bigoplus \to \bigoplus \bigotimes$, usually written as, e.g. $x(a + b)y = xay + xby$, and extended to arbitrary sums and products by induction.

Indeed, if $R$ is an algebra for the monad $\texttt{bag} \circ \texttt{list}$, we have some "double aggregator" $ev : \texttt{bag}(\texttt{list}(R)) \to R$. We can recover $\otimes : \texttt{list}(R) \to R$ by packing our list into a singleton bag, and we can recover $\oplus : \texttt{bag}(R) \to R$ by packing our bag into a singleton list then applying $\lambda$.

# D   Polynomial functors

Polynomial spans are the starting point for our integral transform, but they are also the starting point for *polynomial functors*, which arise in dependent type theory. Let $\mathcal{C}$ be a locally cartesian closed category, and let $\mathcal{C}/A$ denote, for any object $A$ of $\mathcal{C}$, the category of morphisms with target $A$. A polynomial functor starts with a polynomial span:

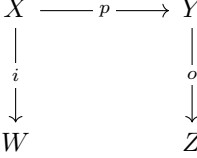

---

[4]Cheng [8] also explains the general problem of composing three or more monads and its relation to the Yang-Baxter equation, which provides further intuition about the unit axioms for semirings.

And it produces a composition of three functors:

$$
\begin{array}{ccc}
\mathcal{C}/X & \xrightarrow{\ \Pi_p\ } & \mathcal{C}/Y \\[2pt]
\uparrow{\scriptstyle i^*} & & \downarrow{\scriptstyle \Sigma_o} \\[2pt]
\mathcal{C}/W & & \mathcal{C}/Z
\end{array}
\tag{11}
$$

Here $\Sigma_o$ and $\Pi_p$ are operations called the *dependent sum* and *dependent product* respectively.

Note that there is a direct correspondence between the three arrows in each of the diagrams 5 and 11. So it is very tempting to ask whether our integral transform is expressible as a polynomial functor. Can our results be rephrased in those terms?

We don't have a complete answer, but we can connect the two pictures, at least in the case of commutative multiplication, via the monoidal category FinPoly, whose objects are finite sets, whose morphisms are polynomial diagrams, and whose monoidal product is given by disjoint union $+$. A result of Tambara says that FinPoly is the Lawvere theory for commutative semirings [25, 17]. What this means is that the strong monoidal functors $F : (\texttt{FinPoly}, +) \to (\texttt{Set}, \times)$ are uniquely determined by giving a commutative semiring structure on the set $F(1)$.

In other words, once we have decided on a commutative semiring structure on $R = [1, R]$, we automatially have $F(V) = F(\sum_V 1) = [1, R]^V = [V, R]$, and the action of $F$ on morphisms can be checked to coincide with our construction of the integral transform.

Likewise, we can interpret finite polynomial functors as the action on the category of categories $F : (\texttt{FinPoly}, +) \to (\texttt{Cat}, \times)$ with $F(1) = \texttt{FinSet}$. Note that $[V, \texttt{FinSet}] = \texttt{FinSet}^V = \texttt{FinSet}/V$, as picking one finite set for each element of $V$ is equivalent to picking a finite set equipped with a function to $V$. So $F$ takes a finite set $V$ to its slice category $\texttt{FinSet}/V$, and likewise takes polynomial diagrams to the associated polynomial functor. In fact, $F$ in this case actually extends to a 2-functor. Since the 2-categorical structure is important for polynomial functors, it may be useful to explore it for integral transforms as well.

In any case, we can see that $[V, \mathbb{N}]$, where $\mathbb{N}$ is the usual natural numbers with addition and multiplication, is just a decategorified version of $\texttt{FinSet}/V$, obtained by considering only cardinalities. Indeed, the existence of such a "decategorification" for transforms over spans was an early inspiration for our present work. But what about categorifying other semirings?

To replace $\mathbb{N}$ with an arbitrary semiring $R$, we would need to find a way to interpret a function $f : W \to R$ as a classifying morphism for some kind of bundle $E \to W$ in a suitable category of geometric objects over $R$. For the min-plus semiring $R = \mathbb{N}^\infty$, one possibility is to define a category of $R$-schemes, which should be certain types of topological spaces equipped with sheaves of $R$-modules.

We don't know of a place this theory is fully developed, but the spectrum functor from rings to topological spaces is extended to poset-enriched semirings in [12]. And this construction is certainly related to tropical schemes, defined in [18]. For $R = \mathbb{R}$, we can also consider the more familiar category of manifolds, or more generally the category of locally compact Hausdorff spaces.

But do polynomial functors work in categories like this? While polynomial functors were developed in type theory over locally cartesian closed categories–too strong of a condition for interesting topology to occur–[34] has shown that polynomial functors can be defined in any category with pullbacks, as long as the "processor" morphism $p : X \to Y$ satisfies an abstract condition called *exponentiability*. $i$ and $o$ can still be arbitrary morphisms.

For some intuition, we quote two results on exponentiability. [6] shows that the exponentiable morphisms in the category of compact Hausdorff spaces are the local homeomorphisms. And [22] shows that a morphism $R \to S$ of commutative rings gives rise to an exponentiable morphism of affine schemes exactly when $S$ is dualizable as an $R$-module. So exponentiability seems to be strongly linked to covering spaces in classical topology, as well as descent theory in modern algebraic geometry.

Expanding on these ideas is far out of scope for the present work, but we hope it gives a glimpse into the possibilities for future development.

## E   Plots of in-distribution performance on CLRS

For plots that illustrate in-distribution performance of our proposed $V^3$ model, against the non-polynomial ($V^2$) model, please refer to Figure 4 and Table 4. Our findings largely mirror the ones from out-of-distribution—with $V^3$ either matching the performance of the baseline or significantly outperforming it (e.g. on Insertion Sort and Floyd-Warshall). We do note that sometimes, matched performance by the non-polynomial $V^2$ baseline in-distribution can be *misleading*, as it significantly loses out to $V^3$ out of distribution (cf. Table 1). This lines up with predicitons of prior art: in-distribution, many classes of GNNs can properly fit a target function [37], but in order to extrapolate well, the alignment to the target function needs to be stronger, as otherwise the function learnt by the model may be highly nonlinear, and therefore less robust out-of-distribution [38].

## F   Test results for the scaled PGN experiments on CLRS

To supplement the aggregated results provided in Table 2, here we provide the per-task results of our scaled PGN experiment. Table 3 provides, for each of the 27 CLRS algorithms we investigated here, the test (out-of-distribution) performance of the PGN model [29], with both the $V^2$ and $V^3$ variant. In all cases, the models compute 96-dimensional embeddings; for memory considerations, the $V^2$ pipeline computes 128-dimensional latent vectors, the $V^3$ addition computes 16-dimensional latent vectors, and these are then all linearly projected to 96 dimensions and combined. We particularly highlight in Table 3 the *edge-centric* algorithms within this set, to emphasise our gains on them. An algorithm is considered edge-centric if it explicitly requires a prediction (either on the algorithm's output or its intermediate state) over the given graph's edges.

Table 3: Test (out-of-distribution) results of all PGN variants on all 27 algorithms in our scaled up experiments, averaged over 8 seeds. Edge-centric algorithms are highlighted in **blue**. Note that most of the benefits of our proposed $V^3$ architecture occur over the edge-centric tasks.

| Algorithm | $V^2$–PGN | $V^3$–PGN |
|---|---|---|
| Activity Selector | $62.28\% \pm 1.02$ | $\mathbf{63.75\% \pm 1.03}$ |
| Articulation Points | $11.91\% \pm 4.46$ | $\mathbf{14.72\% \pm 3.69}$ |
| Bellman-Ford | $\mathbf{80.05\% \pm 0.87}$ | $77.69\% \pm 0.78$ |
| BFS | $\mathbf{99.97\% \pm 0.02}$ | $99.76\% \pm 0.12$ |
| Binary Search | $\mathbf{26.20\% \pm 2.07}$ | $25.57\% \pm 1.95$ |
| Bridges | $\mathbf{26.02\% \pm 1.68}$ | $25.48\% \pm 1.54$ |
| DAG Shortest Paths | $\mathbf{62.62\% \pm 0.44}$ | $62.43\% \pm 0.82$ |
| DFS | $\mathbf{8.70\% \pm 0.73}$ | $8.16\% \pm 0.95$ |
| Dijkstra | $34.60\% \pm 4.13$ | $\mathbf{37.51\% \pm 4.71}$ |
| Find Maximum Subarray | $48.28\% \pm 1.46$ | $\mathbf{52.58\% \pm 1.20}$ |
| Floyd-Warshall | $8.01\% \pm 1.31$ | $\mathbf{17.31\% \pm 0.92}$ |
| Graham Scan | $37.66\% \pm 1.77$ | $\mathbf{42.08\% \pm 1.57}$ |
| Heapsort | $2.34\% \pm 0.15$ | $\mathbf{4.20\% \pm 0.24}$ |
| Insertion Sort | $12.14\% \pm 0.24$ | $\mathbf{18.99\% \pm 0.98}$ |
| KMP Matcher | $\mathbf{2.44\% \pm 0.11}$ | $1.59\% \pm 0.11$ |
| LCS Length | $52.87\% \pm 2.35$ | $\mathbf{67.24\% \pm 4.93}$ |
| Matrix Chain Order | $70.94\% \pm 1.13$ | $\mathbf{74.61\% \pm 0.92}$ |
| Minimum | $\mathbf{58.92\% \pm 1.82}$ | $56.54\% \pm 1.77$ |
| MST-Kruskal | $\mathbf{43.34\% \pm 5.26}$ | $38.42\% \pm 6.82$ |
| MST-Prim | $29.05\% \pm 3.54$ | $\mathbf{29.86\% \pm 3.78}$ |
| Naïve String Matcher | $\mathbf{2.06\% \pm 0.59}$ | $1.80\% \pm 0.46$ |
| Quickselect | $2.22\% \pm 0.08$ | $\mathbf{2.56\% \pm 0.16}$ |
| Quicksort | $2.45\% \pm 0.09$ | $\mathbf{6.82\% \pm 1.01}$ |
| Segments Intersect | $\mathbf{61.77\% \pm 2.15}$ | $61.24\% \pm 1.99$ |
| Strongly Connected Components | $8.98\% \pm 0.56$ | $\mathbf{11.41\% \pm 2.13}$ |
| Task Scheduling | $84.36\% \pm 1.30$ | $\mathbf{85.18\% \pm 0.63}$ |
| Topological Sort | $\mathbf{12.80\% \pm 0.56}$ | $9.91\% \pm 1.63$ |
| Overall average | $35.30\%$ | $\mathbf{36.94\%}$ |

Table 4: Validation (in-distribution) results of all MPNN-based models on all six algorithms studied, across three random seeds.

| Algorithm | $V^2$–large | $V^3$–large | $V^2$–small | $V^3$–small |
|---|---|---|---|---|
| Dijkstra | $92.03\% \pm 0.46$ | $\mathbf{92.70\% \pm 0.34}$ | $91.46\% \pm 0.53$ | $91.54\% \pm 0.49$ |
| Find Maximum Subarray | $81.98\% \pm 2.51$ | $\mathbf{84.71\% \pm 0.93}$ | $81.91\% \pm 1.99$ | $76.29\% \pm 2.46$ |
| Floyd-Warshall | $79.51\% \pm 0.59$ | $\mathbf{90.02\% \pm 0.32}$ | $78.19\% \pm 0.67$ | $88.99\% \pm 0.47$ |
| Insertion Sort | $87.48\% \pm 1.96$ | $87.97\% \pm 1.86$ | $76.12\% \pm 3.77$ | $\mathbf{88.84\% \pm 1.68}$ |
| Matrix Chain Order | $97.69\% \pm 0.07$ | $\mathbf{97.96\% \pm 0.06}$ | $97.59\% \pm 0.10$ | $97.88\% \pm 0.10$ |
| Optimal BST | $\mathbf{92.42\% \pm 0.24}$ | $91.61\% \pm 0.28$ | $91.80\% \pm 0.46$ | $90.77\% \pm 0.63$ |
| Overall average | $88.52\%$ | $\mathbf{90.83\%}$ | $86.18\%$ | $89.05\%$ |

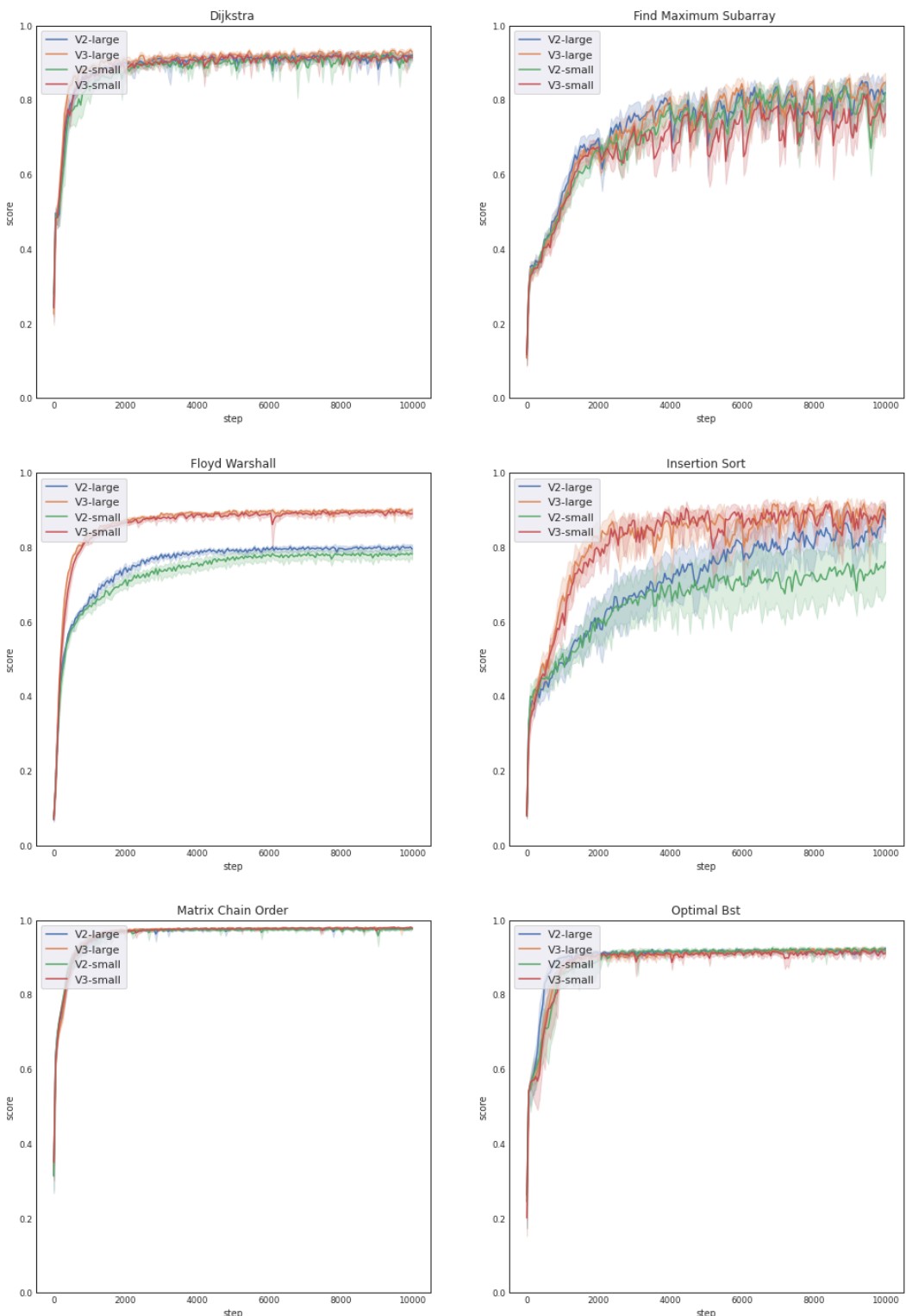

Figure 4: Validation (in-distribution) curves of all models on all six algorithms studied, across three random seeds.