# OpenReview forum: "Graph Neural Networks are Dynamic Programmers"
_NeurIPS.cc/2022/Conference — NeurIPS 2022 Accept_

### Official Review · Reviewer_QbK2 · 2022-06-24

**Rating:** 5
**Confidence:** 2
**Soundness:** 3 good
**Presentation:** 2 fair
**Contribution:** 2 fair

**Summary:**

In this paper, the authors describe the connection between GNNs and dynamic programming via the category theory, specifically, the integral transform. It shows that both GNNs and DPs can be described as a polynomial span diagram and people can improve GNNs' edge updates from the perspective of the polynomial span diagram.

With the analysis from the polynomial span, the authors show that we can use order-3 edge updates to improve the performance of GNNs for edge-centric problems in the CLRS Algorithmic Reasoning Benchmark.


**Questions:**

Can the authors demonstrate the improvement of the order-3 updates with larger GNNs? For example, a 128-dimension embedding and 256-dimension hidden layer?

How's the improved the GNNs compare to the state-of-the-art on the CLRS benchmark? Or, can you improve the state-of-the-art models via the order-3 updates?

**Limitations:**

Though it is an interesting theoretical analysis work for GNNs and DP, the experimental results/settings are not satisfactory enough. (As stated in the weakness.)

**Strengths And Weaknesses:**

Strengths:

This is an interesting and intriguing paper for connecting GNNs with DP as well as elevating the understanding of GNNs. The paper builds the connection between GNNs and DP via the polynomial span diagram from category theory. It also demonstrates a bit the empirical usage of this analysis via the improved edge updates, tested on the CLRS Algorithmic Reasoning Benchmark.

Weakness:

I admit that I'm not an expert in category theory, so I expect a more clear/intuitive explanation of the "integral transform" and the "polynomial span diagram".
The experimental results on the CLRS algorithm reasoning benchmark are done via a very small-scale GNN (hidden size of 8/16), which could not fully demonstrate the practical impact of the order-3 updates. Hence, it doesn't surprise me that higher-order updates improve the small GNN models.

---

> ### Author Response · Authors · 2022-08-02
> **Reply to Reviewer QbK2**
>
> Thank you for your kind remarks and careful suggestions! It is great to see your acknowledgement for our theoretical advances. We’ll provide replies to your questions in turn, and we have updated the paper accordingly.
>
> ### **Polynomial span diagram details**
>
> Thank you for raising this issue, and for the very useful suggestion! We fully agree that a side-by-side plot, linking pseudocode to the integral transform, would be very helpful. We now indeed provide such a Figure (Figure 2); see Appendix A of the paper. We, of course, welcome any and all feedback on improving its clarity!
>
> ### **Additional experimental results (including larger latent spaces & the SOTA model)**
>
> We now, accordingly, provide results for 27 tasks in CLRS (with edge-centric ones highlighted), over the PGN model (which is state-of-the-art on CLRS, when considering per-task win count), computing 96 features (a significant scaling up compared to the original experiments).
>
> The results can be found in Tables 2 and 3. Due to space constraints, these results are partly in Appendix F.
>
> As summarised by Table 2, our proposal improves the PGN model by 4.44% on average for edge-centric tasks, and 1.01% for all other tasks. These benefits are larger and more consistent for edge-centric tasks, as predicted by our theory!
>
> We believe that these experimental findings will further strengthen the paper’s contributions.
>
> ### **Additional context on the polynomial span diagram**
>
> Here is some additional context to help understand the arrows in the polynomial span diagram ($V\leftarrow X\rightarrow Y\rightarrow W$). We welcome any feedback you might have on this!
>
> When we say that data “lives on” a set $S$, we imply that it can be specified using a function $S\rightarrow R$, or as we denote it in the paper, an element of $[S, R]$.  You can think of such a set $S$ as a tensor shape.
>
> Imagine we have data that lives on a set of vertices $V$ (e.g. sender nodes), and we want to perform computations to transform it to data that lives on another set of vertices $W$ (e.g. receiver nodes).
>
> First, in order to obtain data on $W$, we usually need to compute some _messages_ that, once we reduce them, we obtain data living on $W$. The messages’ data lives on a set $Y$ (in the case of GNNs, these are often the edges of the graph—as messages are passed along edges). This specifies the $o : Y\rightarrow W$ morphism of the polynomial span: for each message, we need to know where to send it; i.e., which output $o(y)\in W$ will receive the message given on $y\in Y$.
>
> Then, to actually compute the message, we need to apply a _message function_. This function computes the data that lives on $Y$, but, to do so, it needs access to _arguments_. For this purpose, we define the argument space, $X$, and assume that data living on $X$ specifies all the inputs necessary to compute data that lives on $Y$. This specifies the $p : X\rightarrow Y$ morphism of the polynomial span: for each argument, we need to know which message it is computing; i.e. which message $p(x)\in Y$ will be computed based on the argument $x\in X$.
>
> Finally, where do the arguments for relevant functions come from? They can be seen as all copied from the _inputs_ to our polynomial span, i.e., our original input set $V$. This defines the final morphism in our span: $i : X\rightarrow V$; importantly, it points “the other way” from all other morphisms! This is important: $i(x)\in V$ tells us, for every argument $x\in X$, which input in $V$ its value is copied from.  As a simple example, consider the input function i: {a, b} -> {c}.  Given tabular data like {c: 1}, pulling back this data along i produces the tabular data {a: 1, b: 1}.
>
> Please, let us know if there are any outstanding issues! We look forward to discussing further with you.

---

### Official Review · Reviewer_TCta · 2022-07-12

**Rating:** 6
**Confidence:** 2
**Soundness:** 3 good
**Presentation:** 3 good
**Contribution:** 3 good

**Summary:**

This paper proposes an approach to interpret the operations of both DP and GNNs from the perspective of category theory and abstract algebra. They derive the GNN-DP connection by a diagrammatic abstraction in the computations and recast the algorithmic alignment to align the diagrams of GNNs to some specific algorithm classes. Some experiments on six edge-centric tasks from CLRS Algorithmic Reasoning Benchmark demonstrate the effectiveness of the proposed derive.

**Questions:**

- One major question is that in the evaluation part why use a two-layer MLP with different dimensional sizes, It seems GNNs are discussed in this paper, however, using MLP for evaluation is unclear to me. I am not very familiar with this topic, so please clarify it.
- Furthermore, why choose in-domain and out-of-domain for evaluation, and what is the rationality behind it.

**Limitations:**

The evaluation part is not sufficient.

**Strengths And Weaknesses:**

Strengths:
- This paper theoretically proposes the integral transform to connect GNNs and DP. Furthermore, they provide sufficient theoretical proof to prove the correctness. The theoretical part seems solid to me.
- This paper is good at writing, the target problem is demonstrated clear and the proposed solution is well-discussed.

Weakness:
- The evaluation part is not sufficient and in this paper, only six algorithms are evaluated with different dimension lengths from in-domain and out-of-domain datasets.

---

> ### Author Response · Authors · 2022-08-02
> **Reply to Reviewer TCta**
>
> Thank you for your kind review! We are very happy to hear you found the theoretical results sound. We’ll provide replies to your questions in turn, and we have updated the paper accordingly.
>
> ### **Additional experimental results**
>
> Thank you for this suggestion!
>
> We now, accordingly, provide results for 27 tasks in CLRS (with edge-centric ones highlighted), over the PGN model (which is state-of-the-art on CLRS, when considering per-task win count), computing 96 features (a significant scaling up compared to the original experiments).
>
> The results can be found in Tables 2 and 3. Due to space constraints, these results are partly in Appendix F.
>
> As summarised by Table 2, our proposal improves the PGN model by 4.44% on average for edge-centric tasks, and 1.01% for all other tasks. These benefits are larger and more consistent for edge-centric tasks, as predicted by our theory!
>
> We believe that these experimental findings will further strengthen the paper’s contributions.
>
> ### **On the use of a two-layer MLP**
>
> Thank you for raising this question. We assume you are referring to the “We implemented each of these options using a two-layer MLP…” passage, which we now clarify better in the revised paper.
>
> To clarify the meaning of this passage directly: if one specifies two MLPs (the message function $\psi$ and update function $\phi$ in Equation 1), one completely specifies the dataflow of GNNs.
>
> That is, when we say we use two-layer MLPs, what we meant was that we use GNNs with two-layer MLPs for $\psi$ and $\phi$.
>
> ### **Rationale for out-of-distribution evaluation**
>
> Thank you for enquiring about this. The main focus on neural algorithmic reasoning is to enable robustly learning algorithmic execution with (G)NNs.
>
> This can be done for many purposes: a standard purpose is benchmarking the capability of GNNs to behave algorithmically, but also deployment of such executors on interesting general problems (see Veličković & Blundell, ref. [24] in the paper).
>
> The rationale for why we want to evaluate out-of-distribution is that, in order to probe to what extent has our neural network learnt to truly reason like the underlying algorithm, we need to check how closely aligned it is to the algorithmic rule – and one key property of classical algorithms is giving appropriate results on valid inputs, irregardless of the input distribution.
>
> Even though GNNs are well-aligned to DP, this does not mean that out-of-distribution (OOD) extrapolation is a given – neural networks are typically not designed to extrapolate. Hence, one of the key aims of recent work on neural algorithmic reasoning is to improve OOD performance, and this is the reason why we highlight it in the main paper.
>
> Please, let us know if there are any outstanding issues! We look forward to discussing further with you.

---

### Official Review · Reviewer_wgV4 · 2022-07-20

**Rating:** 6
**Confidence:** 3
**Soundness:** 3 good
**Presentation:** 3 good
**Contribution:** 3 good

**Summary:**

The authors connect GNN and DP by using integral transforms as a common form to represent the operations used in GNN and DP. The common form allows studying GNN and GP in unified framework and authors use the connection to design GNN architecture that better align with the underlying DP algorithm to achieve better empirical performance on benchmark.

**Questions:**

1. I find it difficult to fully grasp details for polynomial span diagram of GNNs. One way this paper can be strengthened is to provide more details on how Diagram (7) and (10) are implemented in practices and what does these abstract operations correspond to in GNN operations (perhaps a side-by-side figure with left being the integral transform diagram, and right being the things like pseudocode)
2. The experiments part can be improved by adding negative examples where V^3 network does not improve over V^2 network for those non-edge-centric CLRS benchmarks, or as authors stated in Line 46-47, by showcasing examples that certain classes of GNN fit certain classes of DP, rather than showing V^3 outperforms V^2 throughout.
3. Can authors explain more on the consistently low performance even for Dijkstra problem (~70%) even though it seems the GNN architecture well matches the DP problem structure here (also for results of other benchmarks with performance as low as ~10%). Can authors provide insights on in what directions performance can be boosted, e.g., for Find Maximum Subarray (currently with only ~10% performance)? The baseline I am mainly referring to are the empirical results in Xu et al. 2019 ICLR. It is hard to persuaded by the title “Graph Neural Networks are Dynamic Programmers” with the these underwhelming experiment performance. Is it because of the limited training data? Perhaps authors can consider downplaying their paper title.
4. Information about experiments setup (including data sample size, hyperparameter tuning steps, etc. that can be replicate the experiments) should be added in line (rather than being put into checklist)

**Limitations:**

See above in "Questions" section

**Strengths And Weaknesses:**

**originality:** On the question being asked, this work is similar to Xu et al. 2019 ICLR but this work is original in that this study connect GNN and DP in a more formal way.

**quality & clarity:** Introduction and background sections are clearly written and motivate this study well. And that dynamic programming (e.g., Bellman-Ford) and GNN be represented in the same framework is well demonstrated. I do not have the expertise on examining all the methods details in section 4 so i defer this to other reviewers. I find the quality of empirical results (section 7) can be improved (see below).

**significance:** this study is important on progress to connect classical DP and GNN and may be used to better understand and propose new type of GNN. However, presentation on methods and empirical results can be improved to strengthen the impact of this paper to broader audience.

---

> ### Author Response · Authors · 2022-08-02
> **Reply to Reviewer wgV4**
>
> We would like to thank you for your careful review, and the positive remarks on our work’s significance! We’ll provide replies to your questions in turn, and we have updated the paper accordingly.
>
> ### **Polynomial span diagram details**
>
> Thank you for raising this issue, and for the very useful suggestion! We fully agree that a side-by-side plot, linking pseudocode to the integral transform, would be very helpful. We now indeed provide such a Figure (Figure 2); see Appendix A of the paper. We, of course, welcome any and all feedback on improving its clarity!
>
> ### **Experimental performance levels**
>
> Thank you for flagging the results on algorithms such as Dijkstra’s.
>
> To comment on these results, we need to make an important note: the results of Xu et al. are given **in-distribution** (that is, the distribution of the test inputs is the same as the distribution of the training inputs), whereas the results of our Table provide results **out-of-distribution** (on 4x larger graphs than ones seen at training time).
>
> Even though GNNs are well-aligned to DP, this does not mean that out-of-distribution (OOD) extrapolation is a given – neural networks are typically not designed to extrapolate. Hence, one of the key aims of recent work on neural algorithmic reasoning is to improve OOD performance, and this is the reason why we highlight it in the main paper.
>
> But, for convenience, we also provided the results of our model in-distribution. You can find these results in Figure 3 (Appendix E) and, as expected, the performance is significantly higher than the OOD regime (and in line with Xu et al.).
>
> ### **Additional experimental results**
>
> Thank you for this suggestion!
>
> We now, accordingly, provide results for 27 tasks in CLRS (with edge-centric ones highlighted), over the PGN model (which is state-of-the-art on CLRS, when considering per-task win count), computing 96 features (a significant scaling up compared to the original experiments).
>
> The results can be found in Tables 2 and 3. Due to space constraints, these results are partly in Appendix F.
>
> As summarised by Table 2, our proposal improves the PGN model by 4.44% on average for edge-centric tasks, and 1.01% for all other tasks. These benefits are larger and more consistent for edge-centric tasks, as predicted by our theory!
>
> We believe that these experimental findings will further strengthen the paper’s contributions.
>
> ### **Experiment setup details**
>
> Thank you for highlighting this – as recommended, we have also copied this discussion from the checklist into the main body of the paper.
>
> ### **Additional context on the polynomial span diagram**
>
> Here is some additional context to help understand the arrows in the polynomial span diagram ($V\leftarrow X\rightarrow Y\rightarrow W$). We welcome any feedback you might have on this!
>
> When we say that data “lives on” a set $S$, we imply that it can be specified using a function $S\rightarrow R$, or as we denote it in the paper, an element of $[S, R]$.  You can think of such a set $S$ as a tensor shape.
>
> Imagine we have data that lives on a set of vertices $V$ (e.g. sender nodes), and we want to perform computations to transform it to data that lives on another set of vertices $W$ (e.g. receiver nodes).
>
> First, in order to obtain data on $W$, we usually need to compute some _messages_ that, once we reduce them, we obtain data living on $W$. The messages’ data lives on a set $Y$ (in the case of GNNs, these are often the edges of the graph—as messages are passed along edges). This specifies the $o : Y\rightarrow W$ morphism of the polynomial span: for each message, we need to know where to send it; i.e., which output $o(y)\in W$ will receive the message given on $y\in Y$.
>
> Then, to actually compute the message, we need to apply a _message function_. This function computes the data that lives on $Y$, but, to do so, it needs access to _arguments_. For this purpose, we define the argument space, $X$, and assume that data living on $X$ specifies all the inputs necessary to compute data that lives on $Y$. This specifies the $p : X\rightarrow Y$ morphism of the polynomial span: for each argument, we need to know which message it is computing; i.e. which message $p(x)\in Y$ will be computed based on the argument $x\in X$.
>
> Finally, where do the arguments for relevant functions come from? They can be seen as all copied from the _inputs_ to our polynomial span, i.e., our original input set $V$. This defines the final morphism in our span: $i : X\rightarrow V$; importantly, it points “the other way” from all other morphisms! This is important: $i(x)\in V$ tells us, for every argument $x\in X$, which input in $V$ its value is copied from.  As a simple example, consider the input function i: {a, b} -> {c}.  Given tabular data like {c: 1}, pulling back this data along i produces the tabular data {a: 1, b: 1}.
>
> Please, let us know if there are any outstanding issues! We look forward to discussing further with you.

---

> > ### Comment · Reviewer_wgV4 · 2022-08-08
> > **Reply**
> >
> > I thank authors for their detailed replies with additional clarity and experimental results.
> >
> > ## Polynomial span diagram details
> > It's great to see authors include Figure 2 to better illustrate the idea of polynomial span. Can the authors add pseudocode for diagram (7) and (10), corresponding to $V^2$ and $V^3$ diagrams. I believe contrasting pseudocode for the two diagrams would add another layer of clarity to readers (especially that the source code is not made available).
> >
> > ## Additional experiment details
> > Thank you for the clarification and these results details provide more clarification and insights in the paper. I have 2 more comments:
> > 1. I think including more discussions on the performance of in-distribution tasks is beneficial: for example, on Dijkstra task, why the performance on V2 and V3 are comparable for in-distribution results and different for out-of-distribution tasks? In addition, can authors include numerical results for these in-distribution tasks?
> > 2. It is indeed interesting and exciting to see that the model improve more for edge-centric tasks compared to all other tasks. However, I do want to comment that this seems not that a clear-cut improvement. I would be curious to know more / or include more discussions about what authors think about these results.
> >
> > P.S. good to add more details to Figure 3 caption (at least add that this is in-distribution results)

---

> > > ### Author Response · Authors · 2022-08-08
> > > **Thank you! Revision uploaded**
> > >
> > > Thank you very much for your kind response and engaging in useful follow-up discussion!
> > > We have just uploaded a new paper revision in line with your comments.
> > >
> > > We wholeheartedly agree that explicitly relating Diagrams 7 and 10 to pseudocode would strengthen the presentation of the paper. We have indeed done so now; please see Figure 3 in Appendix A. (NB: We omitted edge and graph-level input features for clarity, but adding them into the mix should be straightforward.) Let us know if more clarification is needed!
> > >
> > > Regarding your follow-ups on experimental details:
> > >
> > > * [On $V^2$ and $V^3$ performance and generalisation:] Thank you for highlighting this---we now have a more explicit discussion on this when we present validation results in Appendix E. Specifically, these results line up well with predictions of prior art in algorithmic alignment: in-distribution, many classes of GNNs can properly fit a target function (Xu et al., ICLR'20), but in order to extrapolate well, the alignment to the target function needs to be stronger, as otherwise the function learnt by the model may be highly nonlinear, and therefore less robust out-of-distribution (Xu et al., ICLR'21).
> > >
> > > * We will also extract the exact numbers from our validation plots, and make them published in an additional revision tomorrow. We reply now, prior to uploading this revision, to invite any potential follow-up discussion with you, while there is still sufficient time.
> > >
> > > * [Regarding the edge-centric performance improvements:] Thank you for bringing this point to our attention also! Our interpretation, which we hope is useful, would largely rest on the somewhat imprecise manner in which "edge-centric tasks" are selected. Specifically, we select those CLRS problems in which a prediction on the level of the edges is required. But, having to predict something on the edge level does not necessarily mean that edge-centric reasoning is required---it may well be the case that the edge label can be fully derived from the relevant node labels.
> > >   * This is surely the case for tasks like Bridges (where, currently the $V^3$ model is unable to significantly improve on the $V^2$---though their performances are roughly the same).
> > >   * For finding bridges, the ground-truth algorithm computes intermediate results purely over the nodes, then deciding whether to "tag" an edge as a bridge, depending on the values of the relevant two nodes (see, e.g., the algorithm implemented at https://cp-algorithms.com/graph/bridge-searching.html#implementation for one example of this).
> > >
> > > * We have now added more details to the caption of Figure 4 (used to be Figure 3), as you advised.
> > >
> > > Please let us know if any outstanding issues remain! We hope the above addresses your comments appropriately. Your comments certainly helped us strengthen our work further.

---

> > > ### Author Response · Authors · 2022-08-09
> > > **Validation table updated in latest revision**
> > >
> > > Dear Reviewer wgV4,
> > >
> > > Thank you once again for engaging in very fruitful discussion!
> > >
> > > As promised, we just wanted to augment our response from yesterday to note that we have now revised the paper one more time, and included the exact validation (in-distribution) performance values, corresponding to the six validation plots, in tabular form.
> > >
> > > These results are clearly signposted in Table 4 and Appendix E and, while on average they still show an outperformance of the $V^3$ model, all of our comments from the previous reply stand, and we would still prefer out-of-distribution comparisons as a way to measure progress in the area.
> > >
> > > We hope this further improves your view of our paper!
> > > We remain available until the end of the discussion period, should you have any further queries.

---

### Author Response · Authors · 2022-08-02
**Thank you to all reviewers! Revision uploaded**

Dear Reviewers,

We would like to thank you all for your kind and careful comments on our work! We have found them extremely useful.

We have now uploaded a revised paper, which incorporates scaled up experiments (96-dimensional embeddings, PGN model, 27 algorithms from CLRS, across 8 seeds ==> 432 different experimental runs overall), and additional details in the appendices (and otherwise) that aim to improve the overall clarity of our polynomial span construction, and directly relate it to the pseudocode of a typical GNN.

We note that OpenReview suffered some availability issues around the rebuttal deadline, which may have made some of our rebuttal responses accidentally duplicated or deleted. We apologise for any inconvenience caused; please let us know if any of the responses are not visible!

---

### Author Response · Authors · 2022-08-08
**Discussion period follow-up**

Dear Reviewers,

Thank you, once again, for all your reviewing efforts, and especially for the very useful comments.

We just wanted to send a gentle reminder that our ability to communicate with each other ends tomorrow. We believe we have addressed all of your queries, and we'd appreciate any further thoughts you might have.

We would also like to remark, one more time, that OpenReview had suffered some availability issues around the first rebuttal deadline -- therefore, to confirm: currently, there should be one rebuttal response per reviewer, and a global rebuttal reply, summarising our revision. If this is not the case, could you please notify us immediately, so we can resend them?

Thank you very much!

---

### Meta-Review · Area_Chair_JafB · 2022-08-26

**Recommendation:** Accept
**Confidence:** Certain

**Metareview:**

All the reviewers and the AC believe this paper makes solid contributions to GNN and its connection with DP. Therefore, the AC recommends acceptance.

**Award:**

No

---

### Decision · Program_Chairs · 2022-09-14

Accept